# On the Origin and Propagation of the COVID-19 Outbreak in the Italian Province of Trento, a Tourist Region of Northern Italy

**DOI:** 10.3390/v14030580

**Published:** 2022-03-11

**Authors:** Luca Bianco, Mirko Moser, Andrea Silverj, Diego Micheletti, Giovanni Lorenzin, Lucia Collini, Mattia Barbareschi, Paolo Lanzafame, Nicola Segata, Massimo Pindo, Pietro Franceschi, Omar Rota-Stabelli, Annapaola Rizzoli, Paolo Fontana, Claudio Donati

**Affiliations:** 1Research and Innovation Centre, Fondazione Edmund Mach, Via Mach 1, 38098 San Michele all’Adige, Italy; luca.bianco@fmach.it (L.B.); mirko.moser@fmach.it (M.M.); andrea.silverj@unitn.it (A.S.); diego.micheletti@fmach.it (D.M.); massimo.pindo@fmach.it (M.P.); pietro.franceschi@fmach.it (P.F.); omar.rota@fmach.it (O.R.-S.); annapaola.rizzoli@fmach.it (A.R.); claudio.donati@fmach.it (C.D.); 2Centro Alimentazione, Agricoltura e Ambiente, Università degli Studi di Trento, 38098 San Michele all’Adige, Italy; 3Department CIBIO, University of Trento, 38122 Trento, Italy; nicola.segata@unitn.it; 4Laboratory of Microbiology and Virology, Country Health Service APSS, S. Chiara Hospital, 38122 Trento, Italy; giovanni.lorenzin@apss.tn.it (G.L.); lucia.collini@apss.tn.it (L.C.); mattia.barbareschi@apss.tn.it (M.B.); paolo.lanzafame@apss.tn.it (P.L.); 5Emerging Bacterial Pathogens Unit, Division of Immunology, Transplantation and Infectious Diseases, IRCCS San Raffaele Scientific Institute, 20132 Milan, Italy

**Keywords:** SARS-CoV-2, genome, transmission

## Abstract

Background: Trentino is an Italian province with a tourism-based economy, bordering the regions of Lombardy and Veneto, where the two earliest and largest outbreaks of COVID-19 occurred in Italy. The earliest cases in Trentino were reported in the first week of March 2020, with most of the cases occurring in the winter sport areas in the Dolomites mountain range. The number of reported cases decreased over the summer months and was followed by a second wave in the autumn and winter of 2020. Methods: we performed high-coverage Oxford Nanopore sequencing of 253 positive SARS-CoV-2 swabs collected in Trentino between March and December 2020. Results: in this work, we analyzed genome sequences to trace the routes through which the virus entered the area, and assessed whether the autumnal resurgence could be attributed to lineages persisting undetected during summer, or as a consequence of new introductions. Conclusions: Comparing the draft genomes analyzed with a large selection of European sequences retrieved from GISAID we found that multiple introductions of the virus occurred at the early stage of the epidemics; the two epidemic waves were unrelated; the second wave was due to reintroductions of the virus in summer when traveling restrictions were uplifted.

## 1. Introduction

The first case of infection by the severe acute respiratory syndrome coronavirus 2 (SARS-CoV-2) virus was reported in China [1] in late December 2019. The virus rapidly spread globally, and the first indigenous case of COVID-19 was reported in Italy on 20 February 2020 in the town of Codogno, Lombardy, followed by the first recorded Italian fatality attributed to COVID-19 on 21 February 2020 in Vo’, Veneto. Local authorities imposed a 14 day, strict, lockdown on the Vo’ municipality, while a major outbreak in Northern Italy with Lombardy as its epicenter led to a national lockdown that started on 10 March 2020 and was finally uplifted on 3 June 2020. During this first epidemic wave covering a three-month period between March and June, an estimated 226,970 people were diagnosed with COVID-19 infection in Italy, with 33,704 attributed casualties. The number of infected individuals remained low during summer, and started to rise again during the month of October, leading to a second epidemic wave [2] that quickly surpassed the number of infected individuals observed in March, that peaked on 22 November and extended to autumn and winter 2020.

The province of Trento has a population of 545,425 (http://www.statistica.provincia.tn.it/statistiche/societa/popolazione. accessed on 1 March 2020) and is located within a mountainous area, the Dolomites, of approximately 6207 km^2^, 50% of which covered by woodlands. It borders with the Italian regions of Lombardy and Veneto, where the two earliest and largest outbreaks of SARS-CoV-2 occurred in February/March 2020, and is crossed by the major avenue that connects Italy to Germany and Central Europe through the Brenner pass. The population is distributed in the two cities of Trento and Rovereto, located in the Adige valley, and in five major areas separated by mountain ranges, namely the Garda Lake, the Val Giudicarie/Rendena, the Val di Non/Val di Sole, the Val di Fiemme/Val di Fassa, and the Valsugana. Each of these areas are served by local emergency rooms, with major hospitals in Trento and Rovereto (Appendix A).

Trentino Province is periodically visited by tourists both in winter and summer. In particular, ski resorts were visited regularly before the introduction of travel limitations in winter 2020 by tourists mainly from Northern Italy but also from other areas across Europe. In addition, thanks to the importance of tourism and agriculture in the local economy, a large number of seasonal workers regularly move between the Province of Trento and Central and Eastern Europe. This situation favors the superposition of several well defined and distinguishable potential routes of the epidemic spread, making the Province of Trento an interesting case to understand the role of socio-economic drivers, in particular touristic activity, as sources of introduction and reintroduction of SARS-CoV-2 into a well-defined and relatively small geographical area.

In this study, we investigate the origin of the SARS-CoV-2 variants circulating in the province of Trento during the period March–December 2020 by analyzing the virus sequences obtained from patients sampled in the province. We also assess whether the resurgence of the number of infections observed starting from September 2020 was due to endemically circulating strains derived from the first epidemic wave or, rather, to new introductions that might have occurred in the summer period after the uplifting of travel restrictions. For this purpose, we sequenced 253 samples collected between 2 March 2020 and 29 December 2020, and analyzed them in conjunction with samples collected from all over the world and stored in the GISAID repository.

## 2. Materials and Methods

### 2.1. Selection of Samples

The Azienda Provinciale per i Servizi Sanitari (APSS) is the public agency that manages all the public health services in the Province of Trento. In particular, APSS is the public health body in charge of collecting and processing nasopharyngeal swabs from seven emergency rooms wards geographically covering the whole Province, as well as from other sources, including nursing homes. In order to reduce potential selection biases and collect a sample representative of community transmission, positive swabs collected from the ERs over the whole duration of the epidemics were randomly sampled, balancing the sampling so that the seven geographical areas were equally represented.

The selection of the swabs for sequencing was made based on the sampling date over the seven main hospitals serving the territory: Trento, Rovereto, Arco, Tione, Cles, Cavalese and Borgo Valsugana (Appendix A). Swabs were selected to have a good coverage of the first and second epidemic wave (dates of the two waves can be found in Figure 1a).

A total of 253 samples were selected for sequencing in 8 different runs of Oxford Nanopore MinION. Of these, 100 samples were collected in March and 68 in April, representing 3.78% (168/4441) of the positive samples collected by APSS from the first wave, i.e., before 1 June. This collection is highly representative of the variability and of the onset of the SARS-CoV-2 epidemic first wave. The remaining 85 samples were collected between 7 October and 29 December. The viral load of the samples was determined by qPCR. The Cycle Threshold (Ct) was varying between 13 and 39. All the ancillary (swab execution date, Ct) information is available in Appendix A.

### 2.2. Sequencing

RNA was extracted and purified from the specimens by the Microbiology and Virology Unit of the Santa Chiara Hospital in Trento and delivered in dry ice to the Fondazione Edmund Mach for the sequencing. As a first step, we performed a qRT-PCR analysis on each sample with the RealStar^®^ SARS-CoV-2 RT-PCR Kit 1.0 (Altona Diagnostics, Hamburg, Germany) to measure the SARS-CoV-2 content. This was expressed in terms of the Cycle Threshold (Ct) value at a fluorescence threshold of 0.1 RFU on a Roche LightCycler 480 instrument. Samples were then grouped based on their Ct value and the ARTIC network protocol with the primer pools 1 and 2 (IDT ARTIC nCoV-2019 V3 Panel, 500 rxn) for ONT sequencing, carried out with minor changes. Briefly, samples with a Ct value < 16 were diluted 20 times, samples with 18 < Ct < 20 were diluted 10 times, whereas samples with Ct ≥ 20 were used undiluted. The cDNA synthesis and the PCR were then performed as described in the original protocol, but the number of PCR cycles were set as follows: Ct ≤ 21 and diluted samples: 28 cycles, 21 < Ct ≤ 26:30 cycles, 26 < Ct ≤ 30: 31 Cycles, 30 < Ct ≤ 33:32 cycles, 33 < Ct ≤ 35:34 cycles, Ct > 35: 37 cycles. The annealing/extension temperature was set at 63 °C.

The PCR products obtained with the primer pool 1 and 2 were then joined for each sample, and the protocol was performed as described in the original procedure but using 60 ng for the EndPrep step. Finally, 45 ng of the cleaned pooled Barcoded samples were used for the ONT adapter ligation step. Then, 12 to 15 ng of the final cleaned library were used for the preparation of the mix to load on the MinION R9.4.1 flowcell. The sequencing was performed until reaching a satisfactory output, as described in the following paragraph.

Sequencing reads were processed following the ARTIC network SARS-CoV-2 protocol (https://artic.network/ncov-2019, accessed on 1 March 2020). Basecalling and demultiplexing of the fast5 reads was performed using Guppy software v4.4.1 (ONT, Oxford, UK). The sequencing run was inspected in real time with RAMPART v1.1.0 to decide when to stop it, making sure, where possible, that also the amplicons that are notoriously harder to sequence had sufficient coverage to call the consensus. Finally, the consensus was built by using medaka (https://github.com/nanoporetech/medaka, accessed on 1 March 2020) following the ARTIC network bioinformatic pipeline v.1.1.3 that is available at https://hub.docker.com/r/ontresearch/artic_bioinformatics (accessed on 1 March 2020).

### 2.3. Quality of the Sequenced Genomes

We observed a wide variability of the amplification yield of the different amplicons covering the SARS-CoV-2 genome. The per-sample mean read coverage of the sequenced genomes ranged from 1064 to 6655. We found a correlation between genome coverage and viral load. In particular, as the cycle threshold (Ct) value increased, the efficiency of amplification of several amplicons, e.g., Amplicon 1 (covering positions 30–410), Amplicon 9 (2504–2902), Amplicon 29 (8595–8983), Amplicon 31 (9204–9585), Amplicon 74 (22,262–22,650) and Amplicon 91 (27,446–27,854), decreased (full details are available in Appendix A). The coverage of these regions was, in some cases, insufficient for a reliable call of the consensus sequence and, therefore, originated a string of unassigned bases in the final sequence. The number of unassigned bases in the consensus sequence ranged from 121 to 5382 (sample Ph4_RUN3_NB13 that had a Ct value close to 40). Of the total 253 consensus sequences, 198 contained less than 3.3% of unassigned bases, 110 less than 1% and 6 more than 10% (full details are available in Appendix A).

The consensus generation pipeline also identified mutations compared to the reference genome from Wuhan (MN908947.3) and stored them in variant calling format (vcf) files.

### 2.4. Phylogenetic Analysis

#### 2.4.1. Data Preparation and Retrieval

A first phylogenetic classification of the complete set of 253 sequenced SARS-CoV-2 genomes was carried out by using the command line version of the PANGOLIN software (version 2.2.3; https://github.com/cov-lineages/pangolin, accessed on 15 February 2021) with standard parameters (pangolin input. fa-o output directory). Representative genomes from the GISAID repository were added to this set using a random sampling strategy which took into consideration the Pangolin lineage classification. Specifically, 2 representative sequences were selected (one from the period December 2019/July 2020 and another one from August 2020/January 2021) for each lineage with a number of sequences *n* < 20 (see Appendix A) and one sequence for each month for lineages with *n* > 20, except for the lineage B1 (*n* > 150), for which one sequence per week was considered. We also included 2 sequences for other lineages that were continuously circulating globally at the end of the first wave of the epidemic (A, A.1, A.2, A.3, A.5, B, B.1.1, B.1.1.8, B.1.5, B.2, B.2.1, B.2.2, B.2.5, B.3; see https://github.com/cov-lineages/lineages accessed on 15 February 2021) and 2 sequences from lineages which have been connected to particular health concerns (B.1.1.7, B.1.351, P1; https://cov-lineages.org/global_report.html, accessed on 15 February 2021). Finally, 2 of the most ancient sequences from the first reports of the SARS-CoV-2 outbreak were included (EPI_ISL_402119, EPI_ISL_402124). This dataset was composed of a total of 386 sequences, including all the new genomes from Trentino and the sequences sampled from GISAID.

Similarly, a second sequence subsampling was performed on the complete GISAID dataset (see paragraph Minimum Spanning Networks below), with the aim of better characterizing the connections of the Trentino outbreak with the first Italian spread of SARS-CoV-2 which occurred (mainly) in Lombardy. All the Italian sequences present in GISAID as of May 2020 were downloaded, while two representative sequences were randomly selected from a set of non-European countries (Australia, Brazil, Japan, South Africa, USA), sampling one sequence from December/January/February and another one from February/March/April. Three sequences were chosen for each country belonging to a representative group of 10 European nations with geographical proximity or strong commercial links with Italy (Austria, Belgium, Denmark, Finland, France, Germany, Spain, Switzerland, Netherlands and the UK). Only the lineage B.1 was taken into consideration in the sampling, as the vast majority of the first sequences of SARS-CoV-2 sampled in Trentino (Ph1) was composed of this lineage (61/72 B.1, 4/72 B.1.1, 7/72 B.1.1.29). An exception was made for some European countries (Austria, France, Germany, and Switzerland), for which we included 2 extra sequences each for lineage B.1.1. For lineage B.1.1.29, three sequences from the period March/April were added as representatives. A set of Chinese sequences (*n* = 7) belonging to A, B, and B.1 lineages from the first outbreak were also included as an outgroup. Due to the low number of mutations present in this dataset (306 mutations), we ran Nextclade v0.14.2 [3] to filter sequences for their quality, keeping only sequences with a good overall quality, according to the output of the program. The total number of sequences of this set is 284. A complete acknowledgement table for all genomes from GISAID is included in Appendix A.

#### 2.4.2. Sequence Alignment, Data Cleaning and Homoplasy Analysis

All 253 sequences obtained in this study were manually curated and inspected in BioEdit [4], trimming the low identity (<20%) and the fragmented extremities of both ends. This set of sequences was merged with the full set of 133 sequences from GISAID and aligned with MAFFT v7.475 [5]. Manual curation of the resulting alignment was performed in BioEdit, removing the start and end regions consisting of uncharacterized bases (denoted with N’s). A Python script was then used to check for bases different from A, C, T, G and the character “-”, converting them all to “-” when present. A maximum parsimony phylogeny was obtained with mpboot-avx [6], assessing the support of the nodes by using 1000 bootstrap replicates (-bb 1000). The resulting tree, together with its corresponding multiple sequence alignment, were used as inputs for HomoplasyFinder [7]. The presence of homoplasy was further inspected using TreeTime [8], with the proper options for homoplasy detection. The sites identified by both programs (10 sites) were considered highly homoplastic and were therefore removed from the alignment, which was used for the following steps of the analysis. The same procedures were carried out on the second set of sequences aimed to characterize the Italian outbreak, comprising the first batch of Trentino sequences (*n* = 72) and a set of GISAID (*n* = 212) sequences, removing 6 homoplastic sites.

#### 2.4.3. Bayesian Molecular Clock Analysis

Model selection was carried out on both datasets using ModelFinder [9], implemented in IQ-TREE v2.1.2 [10], with the following options: -T AUTO-mem 30G-m TESTONLY. The best-fit models according to Bayesian information criteria were GTR + F + I (general time-reversible + empirical base frequencies + significant proportion of invariant sites) and GTR + F (general time-reversible + empirical base frequencies), respectively, for the datasets with 386 and 284 sequences. A Bayesian molecular clock analysis was performed in BEAST2.6 [11], using tip dating, GTR + gamma with estimated base frequencies, a substitution model with gamma = 4, and a coalescent constant prior. We employed a strict clock, as other authors did in similar studies [12,13], a choice which is further justified by the fact that using a relaxed clock model is more computationally expensive without a relevant increase in statistical fit [14]. A total of 5 independent chains were run with length = 50,000,000, sampling 2500 trees each. The chains were merged together with LogCombiner, subsampling 4500 states each with a 10% burn-in. The convergence of the combined chain was assessed in Tracer (ESS > 200). A maximum clade credibility tree was generated from 10,000 trees using Treeannotator, discarding the first 10% (1000 trees). The resulting phylogeny was inspected with FigTree v1.3.1 (http://tree.bio.ed.ac.uk/software/figtree/) to check for errors and consistency. For the dataset with 386 sequences a second phylogeny, using the same parameters, was reconstructed by adding a root prior within group B (in accordance with [15]). The final results have been inspected in FigTree v1.3.1 and then annotated and visualized using the ggtree R package [16].

### 2.5. Minimum Spanning Networks

#### 2.5.1. Selection of Sequences from GISAID

Genome sequences present in the GISAID database were downloaded on 20 January 2021. The following filters were enabled and performed through the interface of the GISAID website (www.gisaid.org): complete, low coverage excluded and high coverage. Moreover, only entries having “human” as host were selected. These filters yielded a total of 317,057 sequences.

#### 2.5.2. Construction of the Network

The genome sequences of SARS-CoV-2 sampled in Trentino, divided by lineage, were clustered together by using cd-hit-est-2d [17] and compared with the sequences downloaded from the GISAID database as specified above. The first 20 most similar genomes to some sequences from Trentino from every cluster were used as inputs for MAFFT [5] to produce a multiple alignment, using the Wuhan genome (MN908947.3) as the reference. For each Pangolin lineage, a distance matrix was built from the multiple alignment and used as input for the reconstruction of a minimum spanning network, where nodes represent the unique sequences and the links are those that minimize the total distance (measured by the number of mutations between connected sequences) spanned by the network. In a maximum parsimony framework, the minimum spanning network of a set of sequences represents the path that minimizes the number of mutations required to reproduce the observed data: two nodes are connected only if the mutations of the first one are also present in the second.

We then mapped the metadata information (i.e., sampling date and location) on the network to identify possible entry and exit points of the virus in the Trentino area. We defined a possible entrance or exit as a node containing sequences from Trentino but also non-Trentino sequences that share the same set of SNPs. To label a node as ‘entrance’ or ‘exit’, the sampling date of all the sequences within that node was considered. In particular, if the sampling date of the sequences sampled in Trentino was later than that of the other GISAID sequences, then the node was considered as an entrance node. Conversely, if the sampling date of the sequences sampled in Trentino was earlier than that of the other sequences, the node was considered an exit. Moreover, a node containing only non-Trentino sequences that: (i) were connected to a node containing only Trentino sequences; and (ii) contains sequences that have been sampled before the Trentino genomes in the connected node, was also considered a possible entrance. Finally, a node containing only non-Trentino sequences that: (i) were connected to a node with only Trentino sequences; and (ii) contains sequences that have been sampled after the Trentino genomes in the connected node was also considered a possible exit. The minimum spanning networks were pruned to remove nodes not directly connected to a node containing sequences from Trentino.

### 2.6. Mobility Data Processing

Mobility data obtained through mobile phones tracking is available at https://www.google.com/COVID19/mobility/. Data of the Trentino/South Tyrol region for the year 2020 was downloaded and further processed to retain only the information corresponding to the Trentino area. This data reports the variation, compared to a baseline (details are reported at the website https://support.google.com/COVID19-mobility/answer/9824897?hl=en&ref_topic=9822927), of the mobility organized in several categories (e.g., to groceries, pharmacies, workplaces, etc.). We used the average of the values across all the categories as mobility value.

## 3. Results

The first case of COVID-19 in the Province of Trento was reported on 2 March, followed in a few days by a widespread community transmission of the epidemic, with many of the early cases occurring in the winter sport area in the Dolomites mountain range. One week later, on 10 March, the number of reported COVID-19 cases had already reached 50, and more than doubled (102 reported positives) by March 12 (Department of Civil Protection: https://github.com/pcm-dpc/COVID-19) (Figure 1a). In the following three months, there was a first epidemic wave that peaked in late March and slowly decreased following the national trend. Just under 5000 COVID-19 cases were reported in the Province of Trento until the end of June. The number of reported cases remained low during the summer, and started to grow in early September into a second epidemic wave, which by mid-October had surpassed in size the first one. 

The Azienda Provinciale per i Servizi Sanitari (APSS), who collected and analyzed all nasopharyngeal swabs, has seven emergency rooms distributed over the territory of the Province that serve geographically distinct areas (see map reported in Appendix A). In order to collect samples that are representative of the geographical and temporal variability of the first and second epidemic waves, we selected 253 positive swabs from which we obtained draft genomic sequences of SARS-CoV-2 using the ARTIC network protocol for Oxford Nanopore sequencing (see Methods).

### 3.1. Lineages of the First and Second Wave

A total of 438 distinct nucleotide positions were found mutated in the 253 sequenced viral genomes compared to the reference Wuhan sequence (MN908947.3), with a minimum of 3 mutations per sequence and a maximum of 32 mutations per sequence. As expected (see Figure 1b), the samples collected during the second wave had a higher number of mutations compared to the reference than those collected in the first (16 mutations on average compared to 6 average mutations from the samples collected in the first wave). Some samples collected in autumn featured fewer mutations than expected, but it must be noted that these genomes were characterized by a high number of unassigned bases (Ns) that mask some of the expected mutations (see Figure 1b and Figure 2c). A molecular clock plot (Figure 1b) shows that the sequences collected in this study have evolved with a rate that is comparable with the molecular clock rate measured on the complete sequence set available from the GISAID database. For a detailed description of the most abundant mutations found in the dataset, see Figure 2c.

Using the Pangolin lineage classification scheme [18], we found that the genome sequences from the first epidemic wave could be classified into nine lineages, the most frequent of which was B.1 (Figure 2a, Appendix A), which accounted for 151/253 sequences. Lineages B.1, B.1.1 and B.1.1.29 were all identified in the first week, between 2 March and 10 March (Figure 2b), pointing towards multiple independent introductions of the virus in an early stage of the epidemics. During the second wave, the number of Pangolin lineages increased to 13 despite the smaller number of sequences sampled, documenting the increased variability of the circulating strains. The most frequent lineage from the second wave was lineage B.1.177. Only two lineages present in the first wave, namely B.1 and B.1.1.29, were also found in the second wave (Figure 1c). The genome sequences belonging to the B.1 lineage that were found in autumn have a relevant number of mutations not present in the B.1 sequences collected in spring, suggesting that these might be new introductions of this lineage in the Trentino area, rather than endemic circulation in the summer months in asymptomatic individuals of the virus. In contrast, genome sequences assigned to lineage B.1.1.29 found in autumn accumulated fewer mutations than those belonging to the same lineage sampled before the summer. This close relationship among the isolates of the first and second epidemic wave might suggest endemic circulation of this lineage during the summer. However, this hypothesis is not supported by phylogenetic analysis (Figure 3), as detailed below.

An interesting relationship was found among the lineages present in the two epidemic waves (upper panel of Figure 1c), with mobility data of the Trentino area obtained by mobile phone tracking downloaded from Google (https://www.google.com/Covid19/mobility/) (lower panel of Figure 1c). This comparison confirms that mobility restriction rules, applied during the lockdown, influenced virus propagation [19]. Most of the lineages detected in Trentino in spring were no longer detected after the summer period and several other new lineages were introduced, which might be an effect of the increased mobility. Further evidence of this is explained in the next paragraph.

### 3.2. Phylogenetic Analysis

Phylogenetic analysis identifies distinct lineages in the first and second epidemic wave.

In Figure 3, we present a dated phylogeny of 386 SARS-CoV-2 genomes, including all 253 Trentino samples, plus 133 genomes sampled from GISAID (see Section 2).

Bayesian inference sets the outgroup in clade A, a group that does not include the first sequences sampled in China. As this scenario could be a possible artifact [15], we repeated the analysis by explicitly placing a root prior in the B clade (see Section 2). Overall, the phylogeny is poorly supported at most nodes, in accordance with the known fast radiation of SARS-CoV-2 during the outbreak [20]. In particular, most nodes in the B clade, which dated approximately between October 2019 and January 2020, are unresolved, as supported by posterior probabilities lower than 0.5 (in Figure 3, only supported nodes are identified by a gray or black circle).

According to posterior probabilities, none of the autumn variants (blue) can be linked with confidence to any of the spring variants (red). This is a further indication that the autumn variants derived from different introductions and do not seem to have evolved from putative clusters that had remained dormant and/or unnoticed in the province during the summer; a result that is in accordance with what has been recently noted by Hodcroft and colleagues [21]. Most of the genome sequences of the second wave belong to the B.1.177 lineage, first detected in Spain. This clade is well distinct from all others, indicating that its introduction and differentiation in the late summer of 2020 strongly contributed to trigger the second wave of the epidemic.

In some cases, genomes sampled in Trentino during the first wave of the epidemic cluster with high support (red branches), indicating a likely local specific outbreak. Similarly, we observe highly supported clusters of the autumn’s second wave (blue branches).

We further inferred a phylogenetic tree centered on the early stages of the Trentino outbreak, by sampling only first wave genomes (Appendix A). Low supports at nodes are widespread across the tree, even though some nodes are well supported and two distinct outbreaks can be distinguished. The topology and the dates of our phylogeny suggest that the spread of the virus in Lombardy started in December 2019 (this scenario is supported by a node with pp > 80%), followed by a second and distinct introduction of the virus in the area of Rome that didn’t result in further outbreaks (highlighted in yellow). The GISAID sequence EPI_ISL_417445 from Lombardy is sister to all other Italian sequences not included in the small outbreak in Central Italy (pp > 80%). There are some supported subclades belonging to the Lombardy outbreak and containing sequences from Trentino. In these cases, the sequences from Lombardy have usually originated earlier than those from Trentino. Most other nodes are poorly supported, indicating that sequences from Lombardy and Trentino are highly genetically similar. These patterns are compatible with the possibility that the virus moved from Lombardy to Trentino, giving origin to local outbreaks in the latter region.

Interestingly, the two dated phylogenies that we presented differ substantially in the estimation of the root age. In the tree of Figure 3, the time to the most recent common ancestor (TMRCA) corresponds to August–September 2019, while in the tree in Appendix A the TMRCA is October–November 2019. This discrepancy is likely related to the different taxon sampling and to the further quality control steps that we carried out for the second dataset, in which we only included high quality sequences according to the Nextclade output (see Section 2).

### 3.3. Minimum Spanning Network Analysis

Minimum spanning network analysis suggests that both epidemic waves were due to multiple independent introductions of the virus.

In order to understand the possible routes through which the SARS-CoV-2 virus spread in the Province of Trento during the first and second epidemic wave, and to have a better understanding of how many initial entrances there were in the first wave and if the second epidemic wave could be attributed to SARS-CoV-2 strains from the first wave that circulated undetected in the area, we compared our newly sequenced genomes to a selection of the sequences from the GISAID database (see Section 2). Using these data, we defined a minimum spanning network (MSN) for the most important lineages in order to trace the most probable entrances and exits of the virus in and from the region.

The number of mutations differentiating the two connected genomes (lighter links correspond to a higher number of mutations). Different symbols represent the origin of the samples of each genome: only from Trentino (diamonds), only from outside Trentino (squares), or from both (circles). Nodes are labelled with the location of the first (in terms of sampling date) detection of a genome. Possible entry and exit points of the virus in Trentino are indicated as “IN” and “OUT”.

#### 3.3.1. Lineage B.1

Lineage B.1 was first identified in Europe and was dominant in North Italy during the first epidemic wave [22]. The first B.1 sequence was detected in Trentino on 2 March in an individual from the municipality of Dro (sample Ph1_RUN3_NB01, Figure 4), a small town located north of Garda Lake. In the following weeks, the lineage was identified in many geographically non-contiguous areas, indicating sustained circulation and, possibly, several unrelated introductions from neighboring areas (Appendix A). In total, 151 of the 253 sequenced genomes were classified as B.1 (Appendix A). Forty samples collected starting from 5 March (Figure 4, central node) presented only the four basic variants that define the lineage (A23403G, C241T, C3037T, C14408T), while the sequence of Dro Ph1_RUN3_NB01, despite its earlier sampling date, presented the additional mutation C21575T.

A comparison with all the publicly available sequences in the GISAID database on 20 January 2021 (see Section 2) showed that genomes harboring the four basic mutations that define the lineage B.1 were detected early both in Italy as well as in several countries across Europe. In particular, this viral sequence was found in Lombardy (EPI_ISL_486659, collection date: 22 February 2020), Emilia Romagna (EPI_ISL_457699, 22 February 2020), Tuscany (EPI_ISL_738147, 25 February 2020), Wales (EPI_ISL_413555, 27 February 2020), Friuli Venezia Giulia (EPI_ISL_417418, 1 March 2020), Estonia (EPI_ISL_457727) on 1 March 2020, Lazio (EPI_ISL_417921) on 1 March 2020 and North Europe (EPI_ISL_415154, EPI_ISL_488760) respectively, on 1 March 2020 and on 3 March 2020.

A careful analysis of the mutations found in the remaining 111 sequences classified as lineage B.1 in comparison to the sequences available in the GISAID database evidenced that these isolates, despite being closely related both temporally and geographically, display a level of variability that is unlikely due to local evolution, and might in fact point to more than one independent introduction. From an analysis of the MSN (see Section 2), it is possible to identify at least eight possible entry points (marked as “IN” in Figure 4) of this lineage in the Trentino region (Riva del Garda, Trento. Unknown, Arco, Borgo Valsugana, Vermiglio and Novella), supported by the fact that genomes with exactly the same mutations were detected in samples collected outside the Trentino area at an earlier date. Four additional entrances are possible (Faroe Islands/France, Veneto, Germany, Austria see Figure 4), but in these cases, the number of mutations differentiating in sequences from Trentino and their putative ancestors is high, and therefore it is rather difficult to trace the real origin of the virus with sufficient accuracy.

Using the MSN, we could also identify possible exit points of the virus from Trentino, i.e., sequences sampled in Europe that have one earlier sequence from Trentino as the most similar sequence (marked as “OUT” in Figure 4, see Section 2). For example, we found an accession in Ukraine (EPI_ISL_512618) collected on 11 July 2020 that is only connected to one sequence of Trento (Ph1_RUN3_NB23, 11 March 2020, differing by 9 mutations); the same occurred for a sequence detected in Belarus (EPI_ISL_754235) on 11 October 2020 that is connected (although with 11 nucleotide differences) to sequence Ph1_RUN1_NB19, sampled on 10 March 2020. The genome Ph1_RUN3_NB01 present in Dro on 2 March 2020 was also found in Campania on 2 April 2020 (EPI_ISL_833537).

Genomes from lineage B.1 were found also during the second wave of the pandemic, raising the question whether these cases derive from local circulation during the summer, or rather to a second introduction from outside the Trentino region. Despite the fact that the lack of sequences sampled in the period between the two epidemic waves hampers the MSN analysis, the phylogenetic analysis (Figure 3) and the MSN (Figure 4) support the thesis of new introductions from outside of the region. Indeed, the B.1 sequences from the second epidemic wave in Trentino formed phylogenetic clades that also contain sequences from outside Trentino, suggesting complex circulation of this variant in the months between the two epidemic waves. Using the MSN, we found that the most similar sequences to Ph4_RUN2_NB10, the first B.1 sequence that was sampled in Trento in the second epidemic wave (10 October 2020, 100% identical to Ph4_RUN2_NB06, collected in Pozzolengo on 22 October 2020), are two sequences from Faroe Islands (EPI_ISL_526941, collected on 14 March 2020) and Denmark (EPI_ISL437672, collected on 28 March 2020) respectively. However, the relatively large genetic (11 nucleotide mutations) and temporal distance do not allow us to draw definitive conclusions concerning the origin of the isolates from Trentino. Another example is sample Ph4_RUN2_NB15, detected in Sant’Orsola Terme on 29 December 2020, whose closest sequence (with 8 nucleotide mutations) was isolated in Switzerland on 9 June 2020 (EPI_ISL_574812).

We also found mutations that were only detected in sequences from Trentino, which suggests local evolution of the virus, but without evidence of transmission outside of the province. Some examples are mutation T9775G found in Borgo Valsugana on 14 April 2020 (Ph2_RUN3_NB20), mutation A6098G detected in Levico Terme on 9 March 2020 (Ph1_RUN2_NB03), and mutation G13213T sequenced in Trento on 1 April 2020 (Ph2_RUN1_NB14),. Interestingly, none of these variants are present in the GISAID database.

#### 3.3.2. Lineage B.1.177

Lineage B.1.177 was first identified in Spain in early summer and quickly became the most frequent in the second epidemic wave [21]. In Trentino, the first B.1.177 sequence was identified on 8 October in samples from Civezzano and other locations near the city of Trento. After that, sequences from the lineage were found in neighboring areas (Appendix A).

We found no genomes in GISAID that shared the same pattern of mutations identified in this lineage in Trentino, probably due to the fact that only very few sequences were sampled during the summer period in Europe, as circulation of the virus remained at sensibly lower levels compared to spring. As for lineage B.1, and also in the case of lineage B.1.177, more than one entrance during the summer could be identified. Based on the analysis of the MSN (Figure 4), the most probable virus entrances were two sequences from Spain. Indeed, the MSN identifies two distinct epidemic clusters that are closely related to two sequences isolated in Spain (EPI_ISL_824418, EPI_ISL_824445) in two different dates, namely 26 June 2020 and 20 July 2020. Sequences identical to the latter were also found in North Europe at later dates. Other possible routes through which this lineage might have propagated in Trentino link to Scotland (EPI_ISL_534671, 17 August 2020) and to the Italian region Basilicata (EPI_ISL_722878, 15 October 2020). The latter genome has also been detected in the Netherlands on a date that is compatible with possible transmission to Trentino. Other interesting cases include one sequence identified on 13 October that was identical to a sequence detected in the UK (EPI_ISL_606722) only two days later. Another entrance may be from Campania, even if the sampling date is four days earlier than that of the sequence sampled in Trentino, but the latter genome has four mutations that are not present in the virus sequenced in Campania.

The MSN also allowed us to identify one example of possible transmission of the virus from Trentino to Veneto, a neighboring region in Italy. Indeed, one sample collected in Veneto on 9 November 2020 (EPI_ISL_747467) featured the same mutations of our sample Ph3_RUN1_NB06 (collected in Baselga_di_Pinè on 8 October 2020) plus one additional mutation (C25916T). No other identical sequences have been identified among the ones downloaded from the GISAID database.

#### 3.3.3. Lineage B.1.1.29

The B.1.1.29 lineage was first identified in Wales (https://cov-lineages.org/lineages/lineage_B.1.1.29.html), but was also present in the same period in very distant geographical locations such as Cape Town, South Africa [23], and Eastern Germany [24], and was one of the dominant lineages in the early stage of the epidemics in Cyprus [25]. We identified 21 sequences classified as B.1.1.29 in our samples. Three possible entrances can be identified for this lineage. The Ph1_RUN2_NB04 sequence sampled in Canazei on 9 March 2020 is identical to one sequence from Rome (EPI_ISL_417922, 28 February 2020) also present in North Europe (EPI_ISL_416732, 3 March 2020). Moreover, one sequence from Denmark (EPI_ISL_452100, isolated on 7 March 2020) is identical to Ph1_RUN1_NB14 found in Malè on 19 March 2020, and a second sequence from Denmark (EPI_ISL_429484, isolated on 24 March 2020) is identical to Ph2_RUN1_NB11 found in Spormaggiore on 1 April 2020.

One sequence sampled in Lavis (Ph4_RUN3_NB09) has the same mutations found in a sequence sampled in Australia. However, the number of unassigned bases (1990 Ns) in the sequence from Lavis is too high to allow us to draw definitive conclusions.

There is no evident sign of virus exit from the Trentino area, as all possible exit paths cannot be uniquely associated to sequences belonging solely to the Trentino area (e.g., Ph1_RUN1_NB14 could be an exit path for some sequences, but this sequence has been seen to be identical to three other samples in the European dataset). Interestingly, this lineage was found also during the second wave of the epidemic, but it is not clear if this is due to the persistence of the virus in the area or to a new introduction.

#### 3.3.4. Other Lineages Detected in Trentino

Other lineages were present in Trentino during 2020 (Figure 1 and Figure 2 and Appendix A), but the number of sequences detected for each of these is small compared to the lineages described above. Lineage B.1.1 (Figure 4) was detected on 11 March 2020 in Vermiglio, and genomes with the same mutations were also found in Germany (EPI_ISL_775960 on 5 March 2020, EPI_ISL_775953 on 9 March 2020, EPI_ISL_775967 on 11 March 2020), Czech Republic (EPI_ISL_471528 on 9 March 2020), and Switzerland (EPI_ISL_466959 on 11 March 2020). This makes it difficult to trace the origin of the entrance of this lineage in Trentino.

The appearance of the lineage B.1.1.1 in Trentino was observed after the summer, during the second wave. In this lineage, the closest sequence to those sampled in Trentino comes from Abruzzo (EPI_ISL_529014, 20 July 2020) and probably propagated in Trentino through multiple locations like Ronchi Valsugana (Ph3_RUN1_NB19), Rovereto (Ph4_RUN1_NB07, Ph4_RUN1_NB11), Vallarsa (Ph4_RUN1_NB02), and Trento (Ph4_RUN1_NB03).

Lineage B.1.367 appeared in Trentino during the second wave in the neighborhoods of Trento (Ph3_RUN1_NB22 and Ph4_RUN2_NB08). The most similar sequences, which differ by four and five mutations, respectively, from the ones sampled in Trentino, have been found in Switzerland and Belgium (EPI_ISL_512016 on 28 July 2020, EPI_ISL_523924 on 13 August 2020, EPI_ISL_581622 on 24 August 2020, EPI_ISL_626248 on 4 September 2020) at an earlier time.

Lineage B.1.1.305 was detected in Trentino after the summer, but it originated from a pool of genomes belonging to lineage B.1.1 and present in Europe between 24 February 2020 (EPI_ISL_460081, detected in Lombardy) and 13 September 2020 (EPI_ISL_602340, found in Russia). The sequence Ph3_RUN1_NB08 (identical to Ph3_RUN1_NB10), sampled in Baselga di Pinè on 10 October 2020, has nine mutations that distinguish it from this pool of genomes, which makes it difficult to precisely trace the path of this lineage towards Trentino.

## 4. Discussion

In this study we sampled, sequenced, and assembled a total of 253 SARS-CoV-2 sequences from nasopharyngeal swabs collected between March and December, 2020. The high number of sequences produced relative to the population combined with the choice to analyze only swabs sampled in emergency room units spread across the area allowed us to draw an accurate picture of the epidemic in Trentino. Monitoring the number of reported cases in 2020, we could define a first epidemic wave, ending before summer, and a second epidemic wave, starting in September with a peak in November 2020, with a dynamic that closely followed the number of cases at the national level. Despite the low overall sequence variability of the virus, the sequences collected during the first epidemic wave belonged to several distinct lineages, supporting multiple independent introductions of the virus in Trentino in the early stage of the epidemics. A comparison of the sequences with those available in GISAID supports the scenario in which the SARS-CoV-2 epidemic developed in the winter months in the province of Trento were caused by lineages introduced from Lombardy. Interestingly, the estimates of our molecular dating of the first outbreak in the Lombardy area (7 December 2019; 95% HPD intervals: 6 November 2020 to 13 January 2020) are consistent with evidence provided by other methods, such as environmental monitoring [26] and analysis of oropharyngeal swab specimens [27]. Indeed, the early sequences from Trentino show a high degree of similarity with those isolated in the same timeframe from Lombardy, where the major COVID-19 outbreak occurred.

It is interesting to note that the first wave started after a winter ski holiday week, which was characterized by an increase in the number of tourists, especially in the ski areas. 

The second wave was characterized by the entrance of new lineages during the summer period. The lineages detected during the first half of 2020 were no longer found after the summer except for lineages B.1.1.29 and B.1, for which analysis of the mutation pattern and phylogenetic analysis suggests a new introduction, at least for the second lineage. This result supports the hypothesis that mobility restrictions were highly effective for the containment of the epidemics, while the less stringent mobility restrictions adopted during summer facilitated virus circulation and contributed to the second epidemic wave during the autumn and winter of 2020.

Indeed, while the dramatic drop in the number of cases registered both at the local and national level during summer could in principle be attributed to seasonal factors, the fact that the sequences isolated during the second wave were genetically unrelated to those from the first wave in Trentino supports the scenario in which the virus had been locally eradicated during spring, and the second wave was due to a second independent epidemic likely due to introductions that occurred over the summer when mobility restrictions were lifted at the continental level.

The results obtained for the province of Trento are in agreement with previous reports that demonstrate the importance of mobility restrictions in controlling the diffusion of the virus [19,28,29]. In fact, mobile phone data show that during the summer period, the mobility in Trentino considerably increased mainly due to the arrival of a high number of tourists in the area, posing the basis for the second epidemic wave observed in autumn.

Our phylogenetic analysis of SARS-CoV-2 data faced most of the difficulties which have already been highlighted in other research papers on the same topic [12,30,31], such as low posterior supports at most of the nodes. These low values depend on the small number of mutations, especially in the genomes from the first phase of the epidemic, reflecting the fast global spread of the virus. The number of mutations from the Wuhan sequence increased in the second wave of the epidemic, indicating expansion of the viral population, with a higher genetic diversity and number of different variants. Indeed, nodes related to the spread of the B.1.177 lineage (the most common in the second wave of the epidemic in Trentino) tend to have higher posterior supports compared to other clades in the tree.

## 5. Conclusions

The analysis of the lineages identified in the sequenced genomes, in the context of a selection of more than 300,000 sequences downloaded from the GISAID database, shows that the initial phase of the epidemic during March 2020 was due to multiple distinct entrances of the virus in the region. We could also identify possible routes through which viral infection propagated from the province of Trento to other European countries, highlighting the effects of human mobility even outside major travel hubs. Moreover, we could establish that the second epidemic wave was unrelated to the spring epidemic wave, and possibly connected with an increased mobility of people over summer. In conclusion, we show that the travel limitations applied during the spring of 2020 were highly effective in controlling the diffusion of the virus, and that lifting the travel bans during summer created an opportunity to propagate to new viral lineages, causing the second epidemic wave that occurred during autumn and winter 2020. These findings highlight the importance of travel restriction policies to contain contagions and limit the spread of new variants [32].

## Figures and Tables

**Figure 1 viruses-14-00580-f001:**
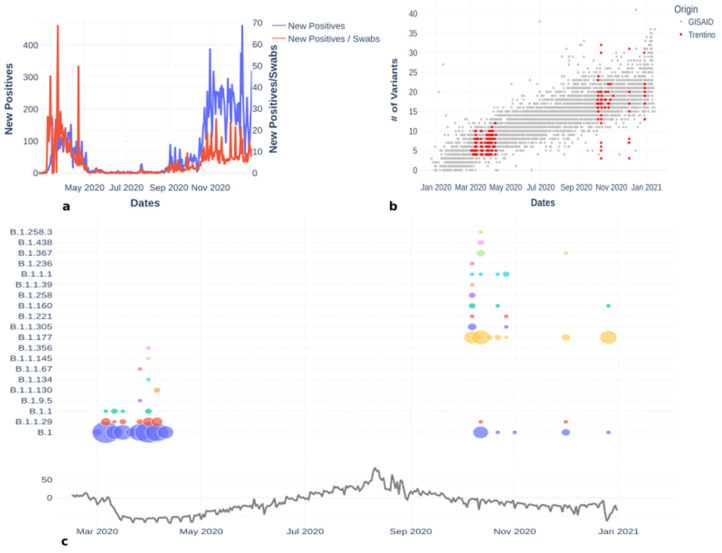
(**a**) The blue line represents the daily new COVID-19 positive cases in Trentino in 2020, while the red line is the percentage of positive daily swabs. The second epidemic wave is characterized by an increase in the number of tests, resulting in a reduction of the ratio between new positive cases and number of tests, but an increase of the absolute number of new positive cases. (**b**) Number of mutations compared to the Wuhan reference genome as a function of sampling date for a selection of sequences from the GISAID database and for the sequences generated in this study (red dots). (**c**) Relationship between the number of sequenced genomes classified according to the Pangolin classification scheme and mobility data monitored in Trentino. The diameter of each circle is proportional to the number of sequenced samples that belong to each lineage.

**Figure 2 viruses-14-00580-f002:**
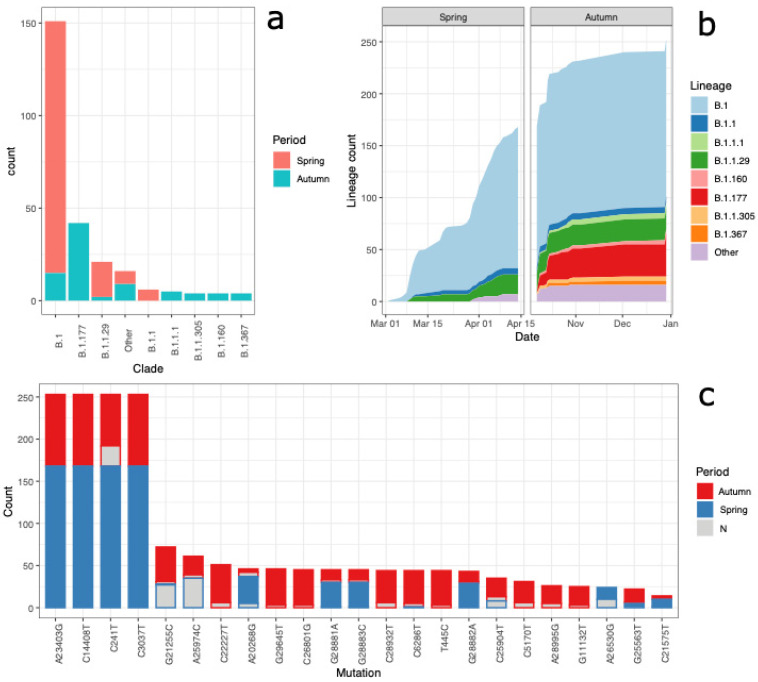
Frequency (**a**) and temporal distribution (**b**) of the Pangolin lineages in Trentino. (**c**) Most common mutations found in the first and second epidemic wave. In gray, we show the sequences where the mutation position was not sequenced. Only mutations present in more than 10 sequences are shown.

**Figure 3 viruses-14-00580-f003:**
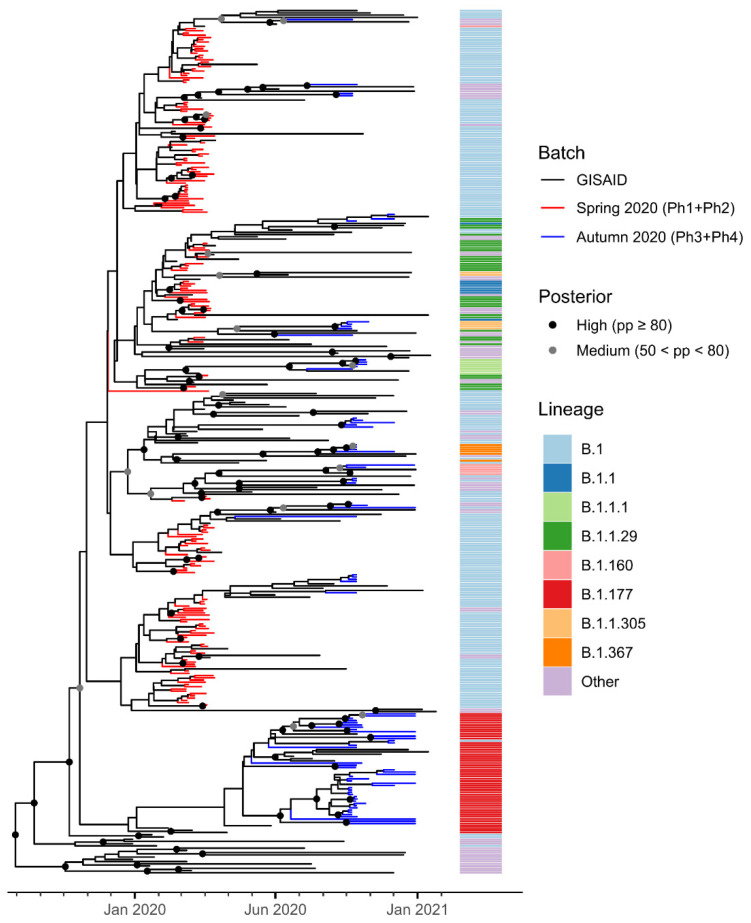
Phylogeny of 386 genomes of SARS-CoV-2, 253 from Trentino and 133 from GISAID. Red branches identify sequences from Trentino from samples collected in the first epidemic wave (spring 2020), while blue branches identify sequences from Trentino from the second epidemic wave (autumn 2020). Black dots mark nodes with high posterior probability (pp ≥ 80%), while gray dots mark nodes with intermediate posterior probability (50% < pp < 80%).

**Figure 4 viruses-14-00580-f004:**
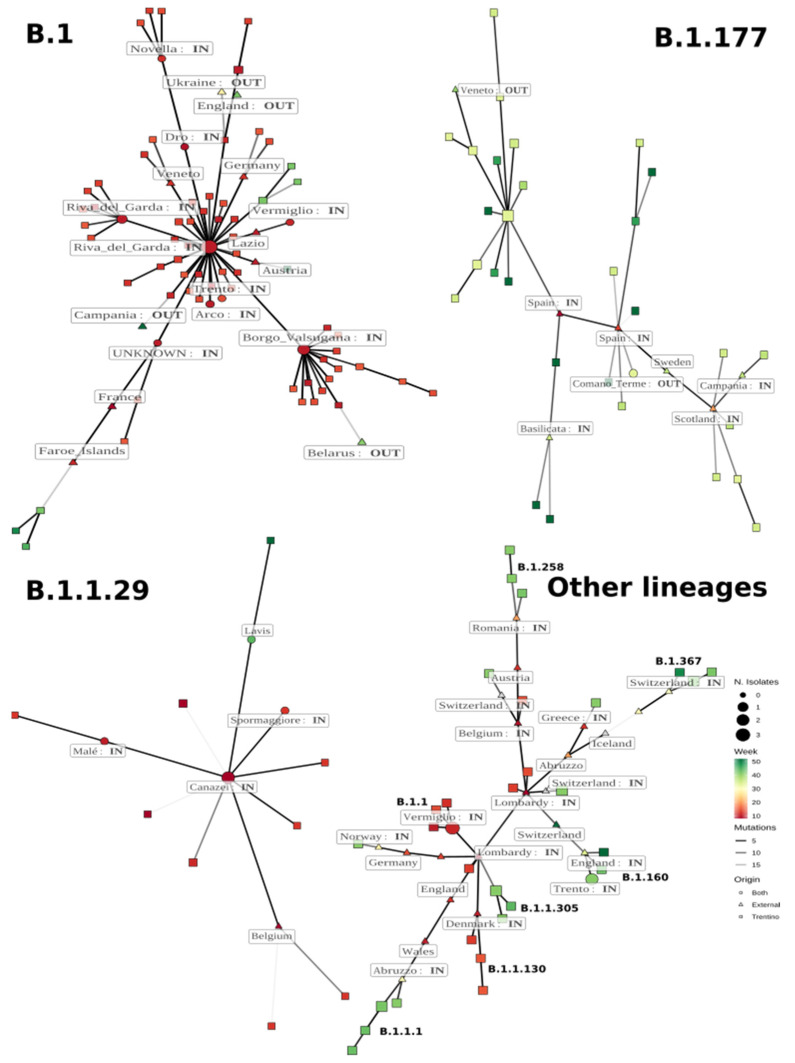
Minimum spanning network of the major lineages sampled from Trentino and other areas across the world. Nodes represent genomes, and the size of the nodes is proportional to the number of samples sharing that sequence. Edges connect two nodes if the most internal node has a subset of mutations from the external one. Colors represent the date of first sampling of each genome, and the transparency of the links is inversely proportional to the number of mutations differentiating the two connected genomes (lighter links correspond to a higher number of mutations). Different symbols represent the origin of the samples of each genome: only from Trentino (diamonds), only from outside Trentino (squares), or from both (circles). Nodes are labelled with the location of the first (in terms of sampling date) detection of a genome. Possible entry and exit points of the virus in Trentino are indicated as “IN” and “OUT”.

## Data Availability

The 253 SARS-CoV-2 sequences produced in this study have been submitted to the GISAID portal (www.gisaid.org) on 26 August 2021. Appendix A reports the correspondence of the GISAID ids of the newly produced sequences with the identifiers reported in this paper.

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
