# Peer review of "On the Origin and Propagation of the COVID-19 Outbreak in the Italian Province of Trento, a Tourist Region of Northern Italy"

_viruses, 2022, doi:10.3390/v14030580_

Round 1
Reviewer 1 Report
no comments
Author Response
We thank the reviewer for taking the time of reading and reviewing our manuscript.
Reviewer 2 Report
This manuscript is very well written, carefully treated and presented. I would recommend to introduce minor changes: to avoid repetitiveness (for example, the description of Trentino appears more than once).
2.4 - OK. 2.5 0 OK. Section 3: may be, the first three paragraphs are unnecessary? At least, they are not results and may be placed in Introduction. 3.1 and 3.2 - OK. 3.3 - minor mishaps. In general, this section is well detailed and clearly presents the results. Section 4 starts with the paragraph that is a description of the results again. In general, I like this section as well. Section 5 starts with unnecessary description again.
Author Response
We would like to thank the reviewer for the useful comments.
We addressed all the points raised by the reviewer and edited the paragraphs accordingly. We enabled the track changes on the manuscript to highlight the places where we performed the suggested corrections.
Reviewer 3 Report
The manuscript is very well written, articulated and really clear.
Even if is not really original as study, it represents a good contribution for understanding the epidemiological evolution of Sars-CoV-2 in province of Trento (Italy) during the first two waves of the epidemic between March and December 2020.
Methods and results are well presented as well discussion and conclusions, also with some interesting insights about the possible routes of the viral transmission to and from Trentino region.
Just a few corrections are necessary:
Line 138: Space between "load" and "In particular"
Line 429: Write in full "i" and "ii"
Line 435: "The" in capital
Line 650: Delete one space between "Trento" and "are"
I think that the manuscript is suitable for publication on Viruses
Author Response

(The authors gave the same response as above.)
